TECHNICAL RELEASE

# svaRetro and svaNUMT: modular packages for annotating retrotransposed transcripts and nuclear integration of mitochondrial DNA in genome sequencing data

Ruining Dong[1,2], Daniel Cameron[1,2,4], Justin Bedo[1,3,†] and Anthony T. Papenfuss[1,2,4,5,*,†]

1 Bioinformatics Division, Walter and Eliza Hall Institute of Medical Research, Parkville, VIC 3052, Australia
2 Department of Medical Biology, University of Melbourne, VIC 3010, Australia
3 School of Computing and Information Systems, University of Melbourne, VIC 3010, Australia
4 Peter MacCallum Cancer Centre, Melbourne, VIC 3000, Australia
5 Sir Peter MacCallum Department of Oncology, University of Melbourne, VIC 3010, Australia

## ABSTRACT

Nuclear integration of mitochondrial genomes and retrocopied transcript insertion are biologically important but often-overlooked aspects of structural variant (SV) annotation. While tools for their detection exist, these typically rely on reanalysis of primary data using specialised detectors rather than leveraging calls from general purpose structural variant callers. Such reanalysis potentially leads to additional computational expense and does not take advantage of advances in general purpose structural variant calling. Here, we present svaRetro and svaNUMT; R packages that provide functions for annotating novel genomic events, such as nonreference retrocopied transcripts and nuclear integration of mitochondrial DNA. The packages were developed to work within the Bioconductor framework. We evaluate the performance of these packages to detect events using simulations and public benchmarking datasets, and annotate processed transcripts in a public structural variant database. svaRetro and svaNUMT provide modular, SV-caller agnostic tools for downstream annotation of structural variant calls.

**Subjects** Software and Workflows, Bioinformatics, Biomedical Science

**Submitted:** 11 February 2022

\* Corresponding author. E-mail: papenfuss@wehi.edu.au

† Contributed equally.

Preprint submitted at https://doi.org/10.1101/2021.08.18.456578

## STATEMENT OF NEED

Structural variants (SVs) are polymorphisms or mutations commonly observed in the genome. They range from simple insertions and deletions to complex chromosomal-scale rearrangements. SVs are a significant source of genomic variability in humans, and SV analysis has rapidly become a part of standard pipelines in genomic studies [1–4]. To call SVs from short-read DNA sequencing data derived from individual samples (e.g., germline or cell lines), matched tumour–normal pairs, or multiple related samples, various tools have been developed [5–7].

Interpretation of SV calls requires additional downstream analyses. For example, in tumour genome analysis, users may want to annotate genes disrupted by SVs, predict

potential in-frame gene fusions [8], classify complex multi-SV events like chromothripsis [9] or chromoplexy [10] and extra-chromosomal DNA detection [11]. However, few tools exist for downstream analysis or annotation of SVs from general purpose SV callers. Much SV annotation software is tightly coupled to specific SV callers (e.g., LINX depends on features of GRIDSS v2) or has its own integrated detector that operates directly on BAM files rather than on an SV call set (e.g., AmpliconArchitect [11] and RetroSeq (RRID:SCR_005133) [12]).

Failure to reuse SV calls from general purpose callers, where these already exist, and reanalysis of alignment files by possibly multiple, specialised callers results in inefficiencies in analysis pipelines. This, in turn, leads to extended run time and higher computing costs. It also misses the opportunity to benefit from improvements in general purpose callers over time. This can be avoided through tools that make appropriate use of SV callsets from general purpose tools. Two biological phenomena currently underserved by annotation tools for general purpose SV calls are Nuclear Mitochondrial insertion (NUMTs) [13] and retroposed transcript (RT) insertion.

The insertion of mitochondrial DNA (mtDNA) fragments into the nuclear DNA during the double-strand break repair has been observed in multiple species, including human and yeast [14–18]. NUMTs are present in the normal genome, having integrated during evolution. Active NUMTs have been observed in cancer; it was proposed that NUMTs are formed during mitosis when the nuclear membrane breaks down, allowing mtDNA to escape from degrading mitochondria, which is accelerated in cancer, and migrate into the nuclear genome [19]. Somatic NUMT events in human cancer cells have not been extensively studied, and further investigation is needed to understand their extent and role in cancer development [19, 20]. Despite their potential biological significance, these events are often overlooked [14]. Dinumt is a NUMT detection tool that identifies NUMTs from whole genome sequencing (WGS) data [13, 21]. Dinumt uses paired reads in which one read maps to mtDNA or known reference NUMTs, and the other read maps to nuclear DNA. Reads that map to nuclear chromosomes are clustered by locations and the insertion sites of the mtDNA sequences are estimated. Read pairs with one read not uniquely mapped are discarded, thus limiting its ability to detect NUMTs in repetitive regions.

RTs are associated with LINE element reactivation in cancer [22], but also occur in the germline leading to processed pseudogenes [23]. RTs can interfere with the expression of their parent genes [24], generate antisense transcripts [25], and compete for microRNA binding with their parent genes [26]. Additionally, mutations introduced by the process may drive cancer evolution, particularly when the RT is inserted into another gene. GRIPper is a tool that identifies possible insertion sites of RTs by looking for discordant paired reads where one read maps to an exon region [27]. The insertion loci are predicted using soft-clipped reads. GRIPper does not look for exon–exon junctions, which are present in most RTs.

Here, we present two R packages for the analysis of SVs calls. svaNUMT and svaRetro provide flexible frameworks to analyse and explore NUMTs and RTs. By using SV callsets generated by general purpose callers, our packages are computationally efficient and avoid reanalysing primary sequencing data. Integrated into the Bioconductor framework, svaNUMT and svaRetro are compatible with many existing tools for further downstream data analysis and applications.

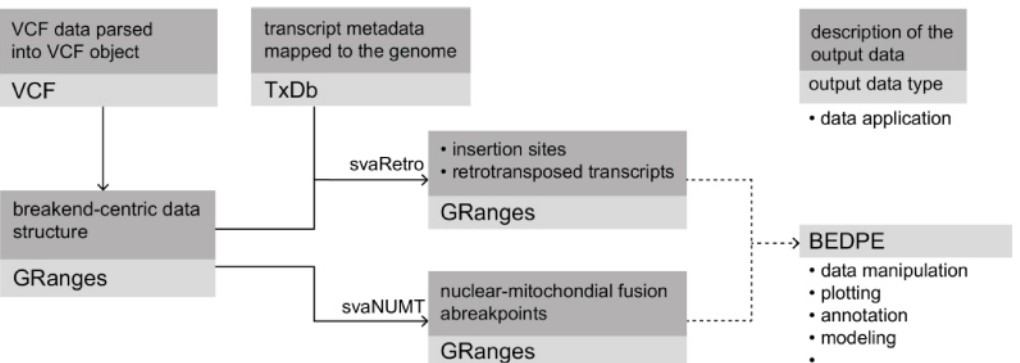

**Figure 1. Workflow of svaRetro and svaNUMT.**
Structural variant (SV) calls are first loaded as variant call format (VCF) objects with VariantAnnotation [28], then converted into breakend-centric GRanges with StructuralVariantAnnotation [29]. svaRetro takes as input the Granges data and a TxDb annotation object, which stores the transcript metadata. The output of svaRetro is a list of GRanges grouped by the source gene of the retrotransposed transcripts. svaNUMT requires only the GRanges object as input. The results are grouped by events and the locations of the breakends. The output can be easily converted to BEDPE format, which is commonly used for downstream analyses.

## IMPLEMENTATION

### Input and output data format

*svaRetro* and *svaNUMT* are designed to work with SV callsets generated by generic callers. In typical use, SV calls in a VCF file are loaded into a breakend-centric GRanges object using VariantAnnotation (RRID:SCR_000074) [28] and StructuralVariantAnnotation (RRID:SCR_018683) [29]. The packages then search for evidence supporting events of interest. For RT detection, svaRetro also requires a TxDb object that stores transcript metadata. The TxDb object can be loaded via pre-existing annotation packages or generated from existing data [30]. The output of the packages are lists of GRanges objects that can be converted to various data formats, including BEDPE, supporting further analysis (Figure 1).

Both svaRetro and svaNUMT take as input a GRanges object with a breakend-centric notation, where a GRanges record is used to represent each breakend, and a breakpoint consists of a pair of breakends. Although breakpoint-centric data structures are available for SV representation (e.g., Pairs object in rtracklayer (RRID:SCR_021325) [31]), we have chosen the breakend-centric notation because it simplifies frequent operations in the analysis, such as overlap finding with genes and repeats. Users can choose threshold parameters before calling svaRetro and svaNUMT to increase sensitivity or precision. Details of each parameter are also described in the documentation of the R packages.

The output formats of svaRetro and svaNUMT are GRanges to support flexible downstream analyses. In cases where alternative formats are required, StructuralVariantAnnotation [29] provides functions for format conversion between BEDPE (RRID:SCR_006646) [32] and Pairs with GRanges. Metadata associated with each input SV call is preserved in the output of svaRetro and svaNUMT, enabling further filtering of the results as required.

### Identifying retrotransposed transcripts

svaRetro identifies RTs using the provided SV calls. RTs are processed transcripts integrated into the genome and characterised by intronic losses and polyadenylation. The candidate

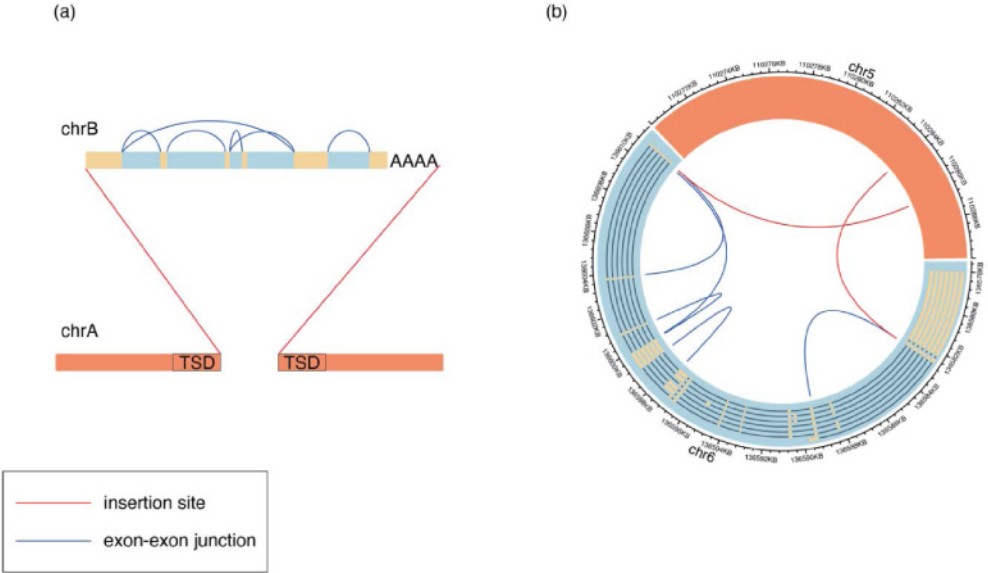

**Figure 2. Breakpoint signatures of retroposed transcript (RT) events.**
(a) A multi-exon RT consists of two breakpoint signatures: exon-exon junctions (blue) and fusions of exon and insertion sites (red). Polyadenylation and target site duplication (TSD) may also be present. (b) A circos diagram (RRID:SCR_002141) [33] of a germline BCLAF1 RT on chr5 detected in sample COLO829 [34] with breakpoint calls of exon-exon junctions (blue) and insertion sites (red). Not all exon–exon junctions were detected by the SV caller, and a transitive call connecting exon 1 and exon 4 is also evident.

insertion sites are scattered across the genome owing to the mobilisation of transposable elements and are frequently combined with target site duplications (TSD). Therefore, except when the transcript comprises only a single exon, an RT should show a signature of intronic deletions – breakpoints aligned with adjacent exon boundaries from the same mRNA transcript. Additionally, the insertion site is detectable as a rearrangement connecting an exonic edge and a second genomic location (see Figure 2).

To detect intronic deletions, overlaps between breakend positions and exon–intron (and intron–exon) boundaries are returned under a maximum gap threshold, denoted by the maxgap parameter. An RT is reported with higher confidence when more exons are present. This quality, denoted by minscore, is evaluated using the proportion of intronic deletions detected from the total possible in the transcript. Depending on the resolution of the SV caller, small exons (e.g., shorter than the read length) can be missed or captured in transitive breakpoints (i.e., a pair of adjacent rearranged segments A ↔ B and B ↔ C are detected as A ↔ C). Meanwhile, breakpoints comprising an exon boundary and a second genomic location are potential insertion sites. Frequent 5′ truncation in retrotransposons means the maximum gap threshold does not apply here. Although events with more supporting breakpoints of exon–exon junctions are more likely to be true events, not all junctions can be called by the upstream general purpose SV caller (e.g., small exon sizes or imprecise breakpoint loci). Further, transcripts with fewer exons may be missed if the threshold is set high. Thus, the parameters should be adjusted according to the purpose of the detection.

The output is a list of GRanges objects consisting of breakend-centric SV calls grouped by the source gene of the retroposed transcript. Each grouped event contains candidate

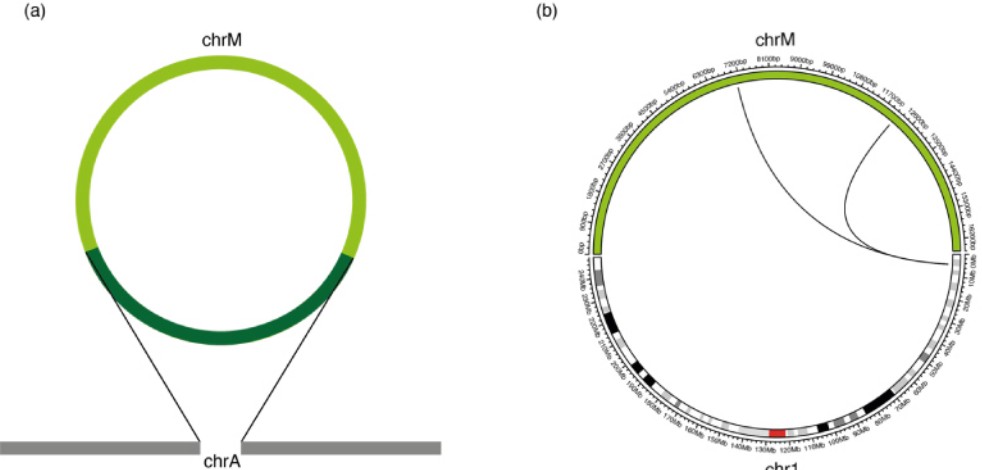

**Figure 3. Breakpoint signatures of NUMTs.**
(a) Schematic of a NUMT event where a sequence from chrM (dark green) is inserted into a nuclear chromosome (chrA). The sequencing read features consist of onebreakpoint connecting chrA and chrM for each insertion site.
(b) A circos diagram [33] of a NUMT event was detected in chr1 of the NUMT simulation.

insertion sites and exon–exon junctions, if available. Each candidate insertion site is annotated by the potential source transcript(s) and whether exon–exon junctions are detected for the source transcript(s). Exon–exon junction calls are annotated by the exon indices, corresponding transcripts satisfying the minscore threshold, and National Center for Biotechnology Information (NCBI) gene symbols.

RT insertion sites can be discovered on both 5′ and 3′ sides, only one side, or none. An insertion site could be missed, even when the breakend is reported in the SV callset, owing to a sizable 5′ truncation despite the tolerant threshold, a 5′ inversion, or a combination of rearrangements.

## Identifying nuclear–mitochondrial genome fusion events

svaNUMT searches for NUMT events by identifying SVs (in breakend notation) supporting the fusion of nuclear chromosome and mitochondrial genome. In the event of mtDNA integration in nuclear genomes, it is expected that split reads and discordant reads are detected near the integration sites. These features, when picked up by an SV caller, are represented as translocation events between mtDNA and nuclear DNA in the SV calls, given that the mitochondrial reference genome is included in the library (see Figure 3). In practice, the inserted mtDNA read sequences may also map to nuclear chromosomes, with many of these sites annotated in known NUMTs [35]. The maximum distance allowed between the insertion sequence and known NUMTs are set by the maxgap_numtS parameter. Alternatively, if the insertion sequence cannot be mapped to a locus with confidence, the NUMT event can be reported in the SV calls as an insertion with a DNA sequence. The min_len parameter sets the minimum length allowed of the insertion sequences. The insertion sequence may not match perfectly with the mitochondria reference genome (chrM), and sequences with an alignment score lower than the min_Align parameter are discarded. svaNUMT can identify NUMTs in all of the above scenarios.

A NUMT event consists of two insertion sites, which can be linked by phasing nearby events. svaNUMT annotates linked insertion sites where possible. The maximum distance



allowed on the reference genome between the paired insertion sites is denoted by the max_ins_dist parameter. Candidate linked nuclear insertion sites are reported by events as a list of GRanges.

## Benchmarking and application

We evaluated svaRetro and svaNUMT on simulated and human cell line data and compared the results with existing tools for RT and NUMT detection. While there were tools developed for transposable elements from WGS data, few tools were available to detect RTs. To the best of our knowledge, GRIPper is the only published RT detection tool [27]. To benchmark NUMT detection, svaNUMT was evaluated against dinumt [13]. For svaRetro and svaNUMT, we used GRIDSS to call SVs on these samples. For the cell lines, results from each tool were compared and discrepancies inspected manually.

### *Testing on simulated data*

We next tested svaRetro and svaNUMT using 500 non-overlapping simulated events on chromosome 1 (chr1). To generate the simulated events, chromosome 1 was first divided into 570 uniform intervals. Of these, 507 overlapped (at least 80% overlap) the set of high-confidence Tier 1 regions defined by Zook *et al.* [36]. Intervals not in high confidence regions were excluded. A final set of 500 intervals was then randomly selected. A transcript sequence, randomly selected from the UCSC annotation database [32], accompanied by a polyadenylation sequence, was inserted into a random location of each interval. Simulated NUMTs were generated through insertions of 500 mtDNA sequences with polyadenylation on the chr1 sequence, where insertion sites were selected using the same method as described above. The mtDNA sequences included 50 each of lengths 10, 20, 50, 100, 200, 500, 1000, 2000, 5000, and 10,000 base pairs (bp). Paired-end reads at 30× mean coverage were simulated using Art (RRID:SCR_006538) [38] using the HiSeq 2500 error profile for both simulated RTs and simulated NUMTs. SV breakpoints of both samples were called by Manta [6] and GRIDSS [5]. We then used svaRetro, GRIPper, svaNUMT and dinumt to detect simulated events and manually inspected the results.

svaRetro detected 470 out of 500 (94%) RTs from the GRIDSS calls and 443 out of 500 (86%) from Manta calls (candidateSV callset). Twenty-three of the 30 undetected events did not have a breakpoint called by GRIDSS within 100 bp of the known insertion sites. Out of the seven events where breakends were detected near the insertion sites, five insertion sequences were not mapped to the reference genome, one event had secondary calls mapped to an alternative locus, and one event was mapped to alternate assembly. Of the 57 undetected events from the Manta calls, 45 did not have a breakpoint called within 100 bp of the insertion sites, 11 insertion sequences were not mapped to the reference, and one inserted transcript was mapped to an alternate assembly. GRIPper detected 295 out of the 500 simulated RTs using the default parameters. All detected RTs were found within 100 bp of the insertion sites. With svaRetro, there was one false positive event detected using the GRIDSS calls, where the insertion site was detected at an alternative mapping position on hs37d5. One false positive was reported using the Manta calls. No false positive was reported using GRIPper. The shortest transcript length detected by GRIPper was 351 bp, while it was 27 bp for svaRetro.

Using GRIDSS SV calls, svaNUMT (using min_len = 8 bps and min.Align = 0.7) detected a total of 440 out of the 500 NUMT events (352 with both breakpoints detected and 88 with

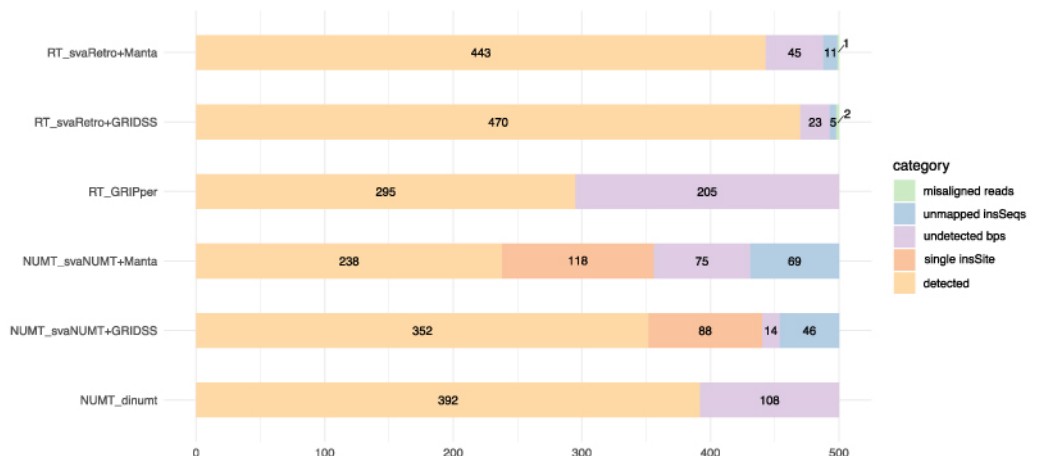

**Figure 4.** **Comparison of simulated retroposed transcripts (RTs) detected using svaRetro and GRIPper, and simulated NUMTs detected using svaNUMT and dinumt.**
Input structural variants (SVs) for svaRetro and svaNUMT were generated with Manta and GRIDSS. For simulated RTs, svaRetro detected 443 (88.6%) from Manta calls and 470 (94%) from GRIDSS calls, and GRIPper detected 295 (59%) from the sequencing reads. The undetected events fell into one of the categories of unmapped insertion sequences (insSeqs), undetected breakpoints (bps), or misaligned reads. For simulated NUMTs, svaNUMT could detect NUMT with both or single insertion sites, including 238 (47.6%) and 118 (23.6%) from Manta calls, and 352 (70.4%) and 88 (17.6%) from GRIDSS calls. dinumt detected 392 (78.4%) NUMTs with both insertion sites. The remainder unreported events either had unmapped insSeqs, or undetected bps. In the online version of this paper this is presented in an interactive form created by the code and frictionless data package presented alongside this work [39, 40]. figure-4.html

one breakpoint only) (Figure 4). Based on Manta SV calls, svaNUMT was able to detect 356 NUMTs (238 with both breakpoints and 118 were with only one). No false positives were reported using the GRIDSS and Manta calls. While there are many possible reasons for the SV callers to miss breakpoints of one of the insertion sites, the presence of polyadenylation could affect the detection of SV breakpoints. Among the undetected NUMT events, some were associated with undetected breakpoints near the insertion sites by the SV callers (14 from GRIDSS and 75 from Manta); some were unmapped insertion sequences (46 from GRIDSS and 69 from Manta) related to short sequence lengths, where most were 10- and 20-bp events. Dinumt detected 392 out of the 500 simulated NUMTs. The minimum length of MT sequence dinumt could detect was 50 bp. No false positives were reported by dinumt.

### Testing on cell line data
We applied svaRetro, GRIPper, svaNUMT and dinumt on a human germline HG002, a GIAB cell line using 60× coverage WGS [41], and a tumour cell line, COLO829, derived from a cutaneous melanoma with the matched lymphoblastoid (normal) cell line [34].

### svaRetro and GRIPper
HG002: GRIPper reports four instances of RTs in HG002, all of which are detected by svaRetro. In addition, svaRetro reports the exon–exon junctions and traces the source of the insertion sequence to the specific transcripts, which is absent in GRIPper (see file in Zenodo [39]).

COLO829: Three RTs were detected by GRIPper in both tumour and matched normal samples. One RT was detected in the normal sample only (see files in Zenodo [39]). For one event detected by GRIPper in both the tumour and the normal samples, the inserted



**Table 1.** Runtime performance of benchmarked methods.

| Variant type | Method | Runtime (min) | |
|---|---|---|---|
| | | Sample: HG002 | Sample: COLO829 |
| NUMT | svaNUMT | 7 | 18 |
| | dinumt | 11 | 77 |
| Retroposed transcript | svaRetro | 62 | 143 |
| | GRIPper | 152 | 227 |

transcript sequence mapped to a processed pseudogene (on chr1) as well as the source gene (on chr16). GRIDSS reported this event as a translocation of the in-reference pseudogene from chr1; therefore, this event was not reported by svaRetro (see file in Zenodo [39]). The rest of the RT events were successfully identified by svaRetro.

### svaNUMT and dinumt

svaNUMT can report NUMTs with different confidence levels that are determined using the quality filters reported in the input SV calls. SV breakpoints labelled with PASS filter are generally considered confident calls. Here, NUMTs detected with both insertion sites labelled PASS, only one insertion site labelled PASS, and no insertion site labelled PASS, are considered of high confidence, intermediate confidence, and low confidence respectively.

HG002: dinumt reported five NUMTs in HG002 (see file in Zenodo [39]), which were all reported by svaNUMT, including one high confidence call, two intermediate confidence calls, and two low confidence calls. svaNUMT also reported five additional NUMTs that were absent in dinumt results, which were all low confidence calls. Upon manual inspection, there was insufficient evidence in the sequencing reads that supported these events.

COLO829: dinumt reported seven instances of NUMTs in the COLO829 tumour and matched normal samples (see files in Zenodo [39]), and all these instances were reported by svaNUMT, including three calls with intermediate confidence and four with low confidence. In addition, six germline low confidence NUMTs were discovered by svaNUMT, but no sufficient evidence was observed in the sequencing reads.

### Runtime

Runtimes of svaNUMT and svaRetro were compared to those of dinumt and GRIPper respectively on HG002 and COLO829 (Table 1). svaNUMT and svaRetro were run using R (version 4.1.3) on an Intel i5 processer and 10 gigabytes (GB) memory. GRIPper was run using four threads.

### Application to gnomAD-SV database

We established a catalog of nonreference RTs using svaRetro on the gnomAD-SV dataset [42], where RTs were largely unannotated. NUMT annotation was not applicable because mitochondrial SVs were excluded from the database. In total, 46,926 candidate insertion sites were detected by svaRetro, including single-exon transcript insertions and/or with insertions with only one side of the insertion detected. The distribution of all source genes and candidate insertion sites are shown in Figure 5. Using the 'PASS' filter, 598 high-confidence insertion sites were supported by exon–exon junctions and high-quality SV breakpoint calls. We classified these high-confidence insertion sites by the repeat class using RepeatMasker [43]. Figure 6 shows that RT insertions can be detected in both nonrepetitive and repetitive sequences.

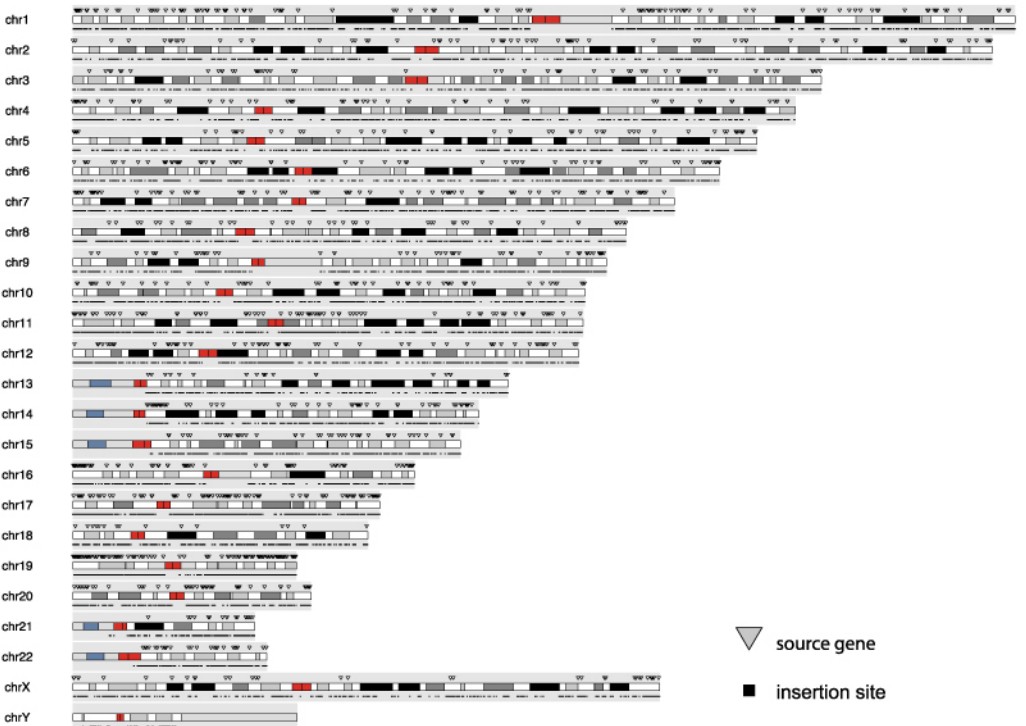

**Figure 5. Landscape of polymorphic retroposed transcripts.**
Chromosomal location of the source gene and insertion locus of retroposed transcripts identified in the gnomAD-SV database using svaRetro.

## CONCLUSION

The R/Bioconductor packages svaRetro and svaNUMT were developed to identify and annotate retrotransposed transcripts and NUMT insertions. Existing annotation tools are often tightly coupled to specific SV callers, or they integrate their own detectors that reanalyse the primary data (alignments). Although some downstream analyses may be unable to avoid such specialised callers, this general approach leads to inefficient use of the computational resources. To address these challenges, svaRetro and svaNUMT provide a modular infrastructure to annotating calls from general purpose SV callers.

A potential shortcoming of this approach is that svaRetro and svaNUMT may be coupled with a less sensitive caller, or biases might be introduced through pipeline choices by the user (e.g., choice of caller, choice of reference, etc.), leading to degraded performance. This is true in all aspects of pipeline design and is mitigated by careful user testing. On the other hand, beyond the previously discussed advantages, both svaNUMT and svaRetro can benefit from technological and methodological improvements in the constituent tools.

svaRetro and svaNUMT demonstrated good performance on simulation and human cell line datasets similar to – or in some instances outperforming – other methods without reanalysis of alignments and the use of specialised detectors. When using low-confidence general purpose SV calls as input, while it can increase the sensitivity of svaRetro and svaNUMT, false positive rates can increase as well. To further demonstrate its capability, novel RT insertions were discovered using svaRetro on a public population SV database.

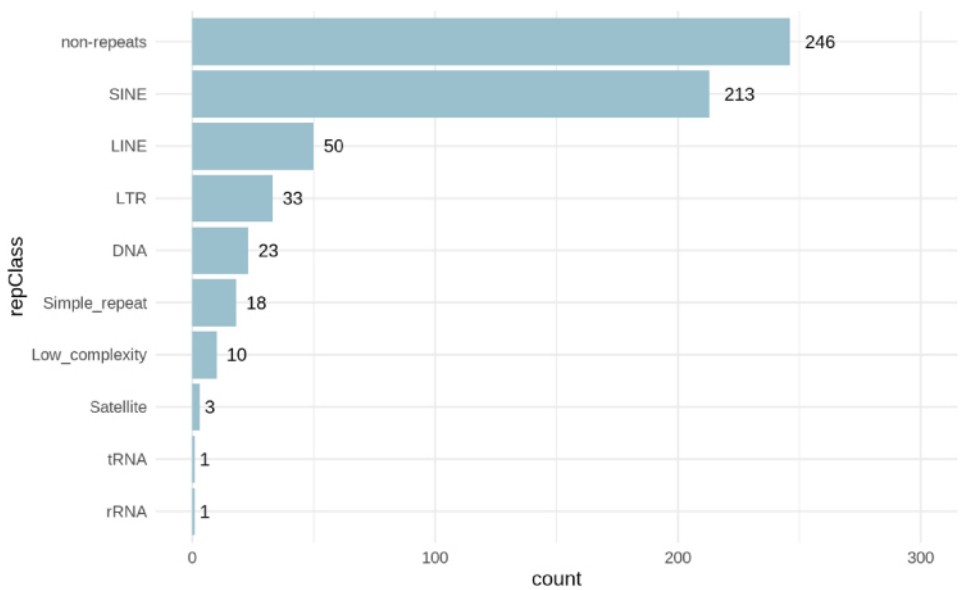

**Figure 6. Genomic context of retroposed transcripts (RTs).**
RepeatMasker [43] repeat class (repClass) annotations of the high confidence RT insertion loci detected from the gnomAD-SV database using svaRetro. 246 (41.1%) of the RT insertion loci were in non-repeat regions. Out of the remainder, the majority were within short interspersed nuclear elements (SINEs) and long interspersed nuclear elements (LINEs), including 213 (35.6%) within SINEs and 50 (8.3%) within LINEs. In the online version of this paper this is presented in an interactive form created by the code and frictionless data package presented alongside this work [40]. figure-6.html

Integrated into the R/Bioconductor framework [44, 45], the packages are compatible with many other available tools for more comprehensive downstream analyses.

## AVAILABILITY OF SOURCE CODE AND REQUIREMENTS

- Project name: svaRetro and svaNUMT
- Project home page:

  – svaRetro: https://github.com/PapenfussLab/svaRetro
  – svaNUMT: https://github.com/PapenfussLab/svaNUMT

- Bioconductor landing pages

  – svaRetro: https://doi.org/doi:10.18129/B9.bioc.svaRetro
  – svaNUMT: https://doi.org/doi:10.18129/B9.bioc.svaNUMT

- Operating systems: Platform independent
- Programming language: R
- Other requirements: R 4.1, Bioconductor 3.14
- License: GPL-3
- RRID:

  – svaRetro: SCR_021380
  – svaNUMT: SCR_021381

## DATA AVAILABILITY

Data and scripts supporting the results of this article are available via the Zenodo repository [39]. The README file includes a more detailed description of each of the files/figure. The Frictionless Data Package and snapshots of the code are also available in the GigaScience Press GigaDB repository [40].

## EDITOR'S NOTE

A CODECHECK certificate for this paper is available confirming that the relevant figures in the paper could be independently reproduced [46]. This was carried out in parallel with generating frictionless data wrappers and using these to create dynamic versions of the figures.

## DECLARATIONS

### List of abbreviations

LINE: long interspersed nuclear element; mtDNA: mitochondrial DNA; NUMT: nuclear-mitochondrial integration; RT: retroposed transcript; SINE: short interspersed nuclear element; SV: structural variant; TSD: target site duplication; WGS: whole genome sequencing

### Ethical approval

Not applicable.

### Consent for publication

Not applicable.

### Competing Interests

The authors declare that they have no competing interests.

### Funding

ATP was supported by an Australian National Health and Medical Research Council (NHMRC) Senior Research Fellowship (grant number 1116955) and the Lorenzo and Pamela Galli Charitable Trust. ATP and DC were supported by an NHMRC Ideas grant (grant number 1188098). JB and ATP were supported by the Stafford Fox Medical Research Foundation. The research benefitted from support from the Victorian State Government Operational Infrastructure Support and Australian Government NHMRC Independent Research Institute Infrastructure Support.

### Author contributions

ATP conceived the study. RD developed the software and wrote the initial draft of the manuscript. DC contributed to software design. JB and ATP oversaw the project. All authors reviewed and contributed to the manuscript. All authors read and approved the final version of the manuscript.

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
