## [Reviewer Report]

Comments on revised manuscriptI feel the authors have addressed my comments. I just have one small comment about their statement in conclusion section line 359-360. They made a statement that “svaRetro and svaNUMT demonstrated good performance on simulation and human cell line datasets similar to - or in some instances outperforming - other methods without re-analysis of alignment and the use of specialized detectors”. While this statement might be all right for simulated data, based on their results in lines 309-319 in cell lines, svaNUMT seems to almost has a 50% false positive annotation rate (although with low confidence). I feel this should be addressed as a caveat in the conclusion and a bit more clearly as false positives in results. Other than that, I do not have any additional comments.

---

## [Reviewer Report]

Reviewer name and names of any other individual's who aided in reviewerSurajit BhattacharyaDo you understand and agree to our policy of having open and named reviews, and having your review included with the published manuscript. (If no, please inform the editor that you cannot review this manuscript.)YesIs the language of sufficient quality?YesPlease add additional comments on language quality to clarify if neededIs there a clear statement of need explaining what problems the software is designed to solve and who the target audience is? NoAdditional CommentsIs the source code available, and has an appropriate Open Source Initiative license <a href="https://opensource.org/licenses" target="_blank">(https://opensource.org/licenses)</a> been assigned to the code?YesAdditional CommentsAs Open Source Software are there guidelines on how to contribute, report issues or seek support on the code?YesAdditional CommentsIs the code executable?YesAdditional CommentsIs installation/deployment sufficiently outlined in the paper and documentation, and does it proceed as outlined?YesAdditional CommentsIs the documentation provided clear and user friendly?YesAdditional CommentsIs there a clearly-stated list of dependencies, and is the core functionality of the software documented to a satisfactory level?YesAdditional CommentsHave any claims of performance been sufficiently tested and compared to other commonly-used packages? YesAdditional CommentsAre there (ideally real world) examples demonstrating use of the software? YesAdditional CommentsIs automated testing used or are there manual steps described so that the functionality of the software can be verified?YesAdditional CommentsAny Additional Overall Comments to the AuthorRecommendationMinor Revisions

---

## [Reviewer Report]

Upload additional filesTRR-202202-01/form/svaRetroandsvaNUMT_review.docxReviewer name and names of any other individual's who aided in reviewerGargi DayamaDo you understand and agree to our policy of having open and named reviews, and having your review included with the published manuscript. (If no, please inform the editor that you cannot review this manuscript.)YesIs the language of sufficient quality?YesPlease add additional comments on language quality to clarify if neededIs there a clear statement of need explaining what problems the software is designed to solve and who the target audience is? YesAdditional CommentsAlthough additional clarification on features of svaRetro can be helpfulIs the source code available, and has an appropriate Open Source Initiative license <a href="https://opensource.org/licenses" target="_blank">(https://opensource.org/licenses)</a> been assigned to the code?YesAdditional CommentsAs Open Source Software are there guidelines on how to contribute, report issues or seek support on the code?YesAdditional CommentsIs the code executable?YesAdditional CommentsIs installation/deployment sufficiently outlined in the paper and documentation, and does it proceed as outlined?YesAdditional CommentsIs the documentation provided clear and user friendly?YesAdditional CommentsIs there a clearly-stated list of dependencies, and is the core functionality of the software documented to a satisfactory level?YesAdditional Comments Additionally, it might be useful to state in description on Github, R version required to install the tool (it doesn’t work with versions older than 4.1)Have any claims of performance been sufficiently tested and compared to other commonly-used packages? NoAdditional Comments1) The authors also need to benchmark their tools against other previously developed tools that they used for comparison (dinumt and GRIPper) using the simulated data.
2) Authors state they found calls that were not found by the other tool. This needs to be further tested to show the results were true positive. In fact, there is no test done to look at the false positives. Therefore, doing a test on their entire results for false positive/ true positive is essential.
Are there (ideally real world) examples demonstrating use of the software? YesAdditional CommentsIs automated testing used or are there manual steps described so that the functionality of the software can be verified?YesAdditional CommentsBut there is a discrepancy for svaNumt. The following command on github “NUMT <- svaNUMT::numtDetect(gr, numtS, genomeMT, max_ins_dist = 20)” doesn’t work. Instead this worked “NUMT <- svaNUMT::numtDetect(gr, max_ins_dist = 20)”
Any Additional Overall Comments to the AuthorRecommendationMajor Revisions